# Zero4D: Training-Free 4D Video Generation From Single Video Using Off-the-Shelf Video Diffusion Models

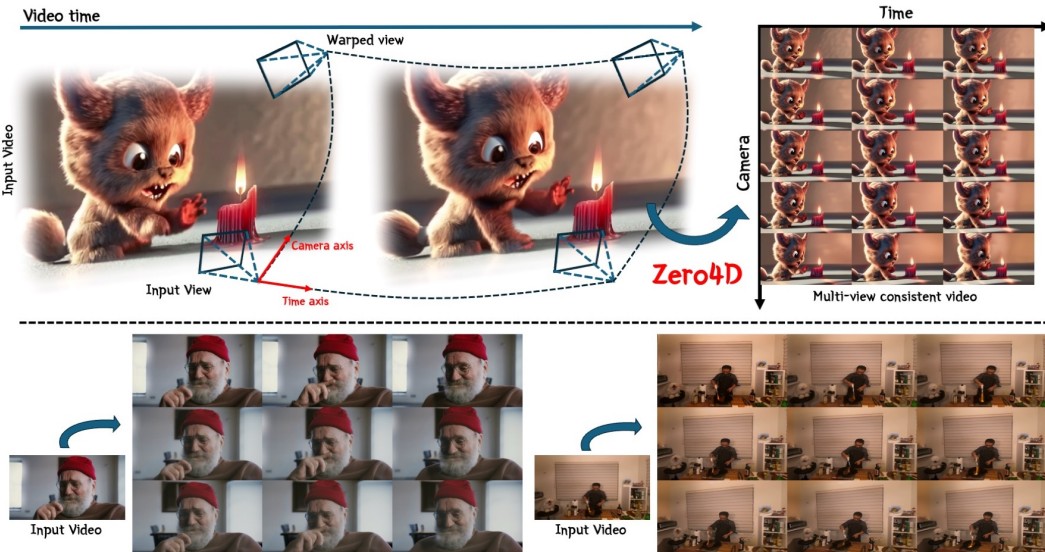

Figure 1: **Zero4D** is a **training-free** multi-view synchronized video generation framework that takes a single monocular video and generates a grid of camera-time consistent frames. It first utilizes a depth estimation model to warp target view frames from the input video (top-left), then repurposes the image-to-video diffusion model to sample multi-view frames synchronized in both camera and temporal dimensions (top-right). Using an off-the-shelf video diffusion model without training, our approach can generate multi-view videos for both synthesized and real-world footage. **Project Page**.

## ABSTRACT

Multi-view and 4D video generation have recently emerged as important topics in generative modeling. However, existing approaches face key limitations: they often require orchestrating multiple video diffusion models with additional training, or involve computationally intensive training of full 4D diffusion models—despite limited availability of real-world 4D datasets. In this work, we propose a novel training-free 4D video generation method that leverages off-the-shelf video diffusion models to synthesize multi-view videos from a single input video. Our approach consists of two stages. First, we designate the edge frames in a spatio-temporal sampling grid as key frames and synthesize them using a video diffusion model, guided by depth-based warping to preserve structural and temporal consistency. Second, we interpolate the remaining frames to complete the spatio-temporal grid, again using a video diffusion model to maintain coherence. This two-step framework allows us to extend a single-view video into a multi-view 4D representation along novel camera trajectories, while maintaining spatio-temporal fidelity. Our method is entirely training-free, requires no access to multi-view data, and fully utilizes existing generative video models—offering a practical and effective solution for 4D video generation.

# 1 INTRODUCTION

Since the introduction of the diffusion and foundation models (Ho et al., 2020; Rombach et al., 2021; Xie et al., 2024), 3D reconstruction has advanced significantly, leading to unprecedented progress in representing the real world in 3D models. Combined with generative models, this success drives a renaissance in 3D generation, enabling more diverse and realistic content creation. These advancements extend beyond static scene or object reconstruction and generation, evolving toward dynamic 3D reconstruction and generation that aims to capture the real world. Previous works (Bahmani et al., 2024b; Zeng et al., 2024; Singer et al., 2023; Zhao et al., 2023; Bahmani et al., 2024a) leverage video diffusion models and Score Distillation Sampling (SDS) to enable dynamic 3D generation. However, most existing approaches primarily focus on generating dynamic objects in blank or simplified backgrounds (e.g., text-to-4D generation), leaving the more challenging task of reconstructing or generating real-world scenes from text prompts, reference images, or input videos largely unaddressed. In contrast to the abundance of high-quality datasets for 3D and video tasks, 4D datasets with multiview, temporally synchronized video remain extremely scarce. As a result, a core challenge in training 4D generative models for real-world scenes lies in the lack of comprehensive, large-scale multi-view video datasets. To overcome these limitations, recent works such as 4DiM (Watson et al., 2024) propose a joint training diffusion model with 3D and video with a scarce 4D dataset. CAT4D (Wu et al., 2024) proposes training multi-view video diffusion models by curating a diverse collection of synthetic 4D data, 3D datasets, and monocular video sources. DimensionX (Sun et al., 2024) trains the spatial-temporal diffusion model independently with multiple LoRA, achieving multi-view videos via an additional refinement process. Despite several approaches, the scarcity of high-quality 4D data makes it difficult to generalize to complex real-world scenes and poses fundamental challenges in training large multi-view video models.

To address these challenges, we introduce *Zero4D*—a novel zero-shot framework for 4D video generation. Zero4D generates synchronized multi-view 4D video from a single monocular input video by leveraging an off-the-shelf video diffusion model (Blattmann et al., 2023), without requiring any additional training. Building upon the prior observations (Wang et al., 2024a; Wu et al., 2024) that 4D video is composed of multiple video frames arranged along the spatio-temporal sampling grid (i.e., camera view and time axes), generating a 4D video can be regarded as populating the sampling grid with consistent spatio-temporal frames. Consequently, our approach achieves this through two key steps: (1) We first designate the boundary frames of the spatio-temporal sampling grid as key frames and synthesize them using a video diffusion model. To ensure structural fidelity, we incorporate a depth-based warping technique as guidance, encouraging the generated frames to conform to the underlying scene geometry. (2) We repurpose the interpolation capabilities of a video diffusion model to fill in the remaining frames through bidirectional diffusion sampling, resulting in a fully populated and temporally coherent 4D grid. Throughout both stages, our method enforces spatial and temporal consistency across the entire grid.

Our main contributions can be summarized as follows:

- We propose a novel framework that can generate 4D video from a single video via an off-the-shelf video diffusion model without any training or large-scale datasets. To the best of our knowledge, our approach is the first interpolation based *training-free* method to generate synchronized multi-view video— previously regarded as infeasible.
- This is made possible by a novel synchronization mechanism, which guarantees high-quality outputs while maintaining global spatio-temporal consistency. Specifically, we alternate bidirectional video interpolation across both the camera and temporal axes to align motion and appearance throughout the sequence.
- Our framework outperforms baselines in maintaining global spatio-temporal consistency and demonstrates robust 4D video generation capability, achieving competitive performance across diverse quantitative and qualitative evaluations even without additional training.

# 2 RELATED WORK

**Video generation with camera control.** Several studies try to train a multi-view diffusion model for spatially consistent image generation (Shi et al., 2023; Wang & Shi, 2023; Liu et al., 2023; Kant et al., 2024; Gao et al., 2024; Melas-Kyriazi et al., 2024). ReCapture (Zhang et al., 2024) trains the novel

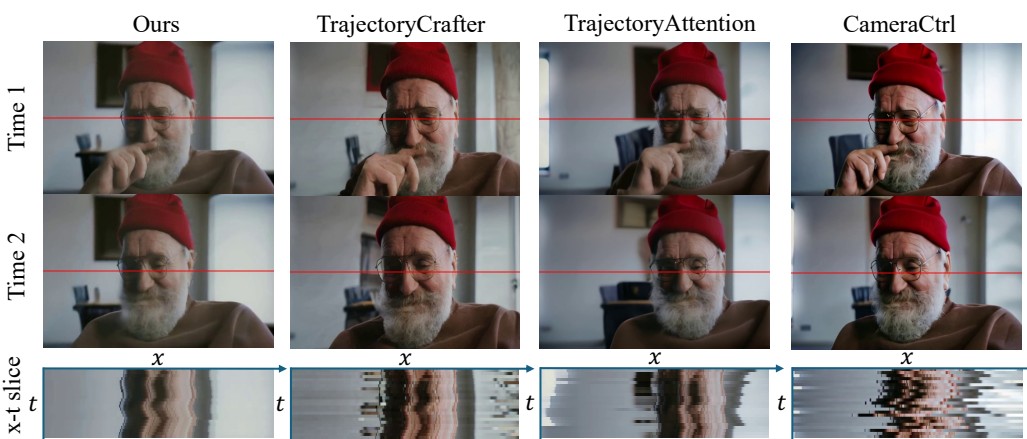

Figure 2: **Qualitative comparison.** We compare our method with baseline models in terms of novel-view video generation and global spatio-temporal consistency. Given a single input video, both baselines and ours generate outputs across multiple views and time steps. To evaluate global consistency, we leverage baselines to produce bullet-time videos at all input frames and re-align them to a fixed viewpoint. We also visualize x–t slices (red lines) to highlight temporal coherence. While baselines exhibit inconsistencies across views and time, our method preserves spatio-temporal coherence and yields high-quality multi-view videos.

Table 1: Comparison of camera-controllable video diffusion models. Unlike prior approaches, Zero4D can generate 4D-consistent videos with camera control without requiring additional training.

| Model | Training-Free | Camera Control | 4D Consistency |
|---|---|---|---|
| Camera Controllable Video Diffusion Model | ✗ | ✓ | ✗ |
| 4D Video Diffusion Model | ✗ | ✓ | ✓ |
| **Zero4D (Ours)** | ✓ | ✓ | ✓ |

camera trajectory video diffusion model from a single reference video with existing scene motion. CameraCtrl (He et al., 2024) proposes a plug-and-play camera module in the video diffusion model to control video generation with precise and smooth camera viewpoints. TrajectoryCrafter (YU et al., 2025) and TrajectoryAttention (Xiao et al., 2025) fine-tune video diffusion models to generate novel-view videos along a given camera trajectory using depth-based warping. These approaches can be categorized as *camera-controllable video diffusion models*. However, although these models can synthesize novel views conditioned on warped videos, they fail to produce 4D-consistent videos that ensure global consistency across multiple views and multiple time steps (see Table 1).

**4D generation.** Recent advancements in 4D generation have been driven by numerous pioneering works exploring various conditioning methods. Several approaches have leveraged score distillation sampling in conjunction with video diffusion models or multi-view image diffusion models to generate 4D content from text prompts (Bahmani et al., 2024b; Zeng et al., 2024; Singer et al., 2023). However, these approaches largely focus on generating dynamic objects in blank backgrounds. A notable example is CAT4D (Wu et al., 2024), which synthesizes 4D videos conditioned on multiple input modalities using a multi-view video model trained on a curated synthetic multi-view dataset. Similarly, Van Hoorick et al. (2024b) introduces a framework for novel-view synthesis of dynamic 4D scenes from a single video. This method is trained on synthetic multi-view video data with corresponding camera poses, enabling high-fidelity 4D reconstructions. Concurrently, Yu et al. (2024) proposes text-to-4D scene generation pipelines that integrate video diffusion models with canonical 3D Gaussian Splatting (3DGS) (Kerbl et al., 2023), ensuring spatio-temporal consistency in the generated 4D outputs. Furthermore, Wang et al. (2024a) enhance video diffusion models by introducing a parallel camera-temporal token stream and a learnable synchronization layer, which effectively fuses independent tokens to maintain camera and temporal consistency across generated frames. While these *4D video diffusion models* enable camera control and maintain multi-view and temporal consistency, they rely on training a large diffusion model with 4D data, which is limited in availability and costly to obtain (see Table 1).

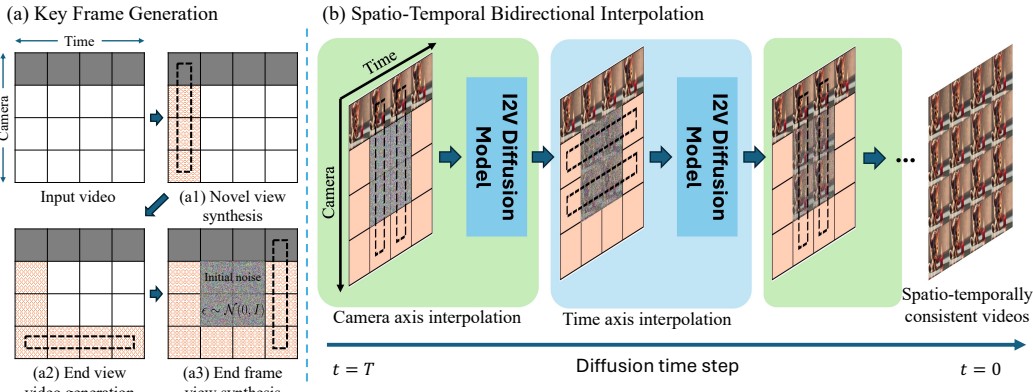

Figure 3: **Generation pipeline of Zero4D: (a) Key frame generation step:** Starting from the input video(shown as the gray-shaded row), we sequentially generate boundary frames—novel view synthesis, end-view video generation, and end-frame view synthesis—where each step leverages the results of the previous one. **(b) Spatio-temporal bidirectional interpolation step:** Starting from the noisy frames, we alternately perform camera-axis and time-axis interpolation, each conditioned on boundary frames, to progressively denoise the 4D grid. Through this bidirectional process, noisy latents are refined into globally coherent spatio-temporal videos.

## 3 ZERO4D

Let $x[i,j] \in \mathbb{R}^{H \times W}, i = 1, \cdots, N, j = 1, \cdots, F$ denotes the image at the $i$-th camera viewpoint and the $j$-th temporal frame, where $H$ and $W$ denote the height and width of the image, respectively (see Fig. 3(a)). Then, the input video captured from a single camera viewpoint $c$ is denoted as $x[c,:]$, whereas the multi-view images at the temporal frame $f$ are represented by $x[:,f]$. The goal of Zero4D is then to populate the spatio-temporal video grid (or camera-time grid) $x[:,:]$ by generating frames across multiple camera poses. The key innovation is that the spatio-temporal grid can be populated entirely at inference time, without any training—a task once thought impossible. As illustrated in Fig. 3, the overall reconstruction pipeline of Zero4D is composed of two steps: 1) key frame generation and 2) spatio-temporal bidirectional interpolation along the time and camera axes in an alternating manner. In this section, we describe each in detail.

### 3.1 KEY FRAME GENERATION

As shown in Fig. 3(a), the key frame generation is achieved through three steps. Specifically, given a input video denoted by $x[1,:]$, we first perform novel-view synthesis, followed by end-view video frame generation. These two steps are achieved through diffusion sampling, guided by warped views. Finally, we complete the rightmost column using diffusion-based interpolation sampling.

**Novel view synthesis (a1).** First, we synthesize novel view video $x[:,1]$ from the first frame image $x[1,1]$ using the I2V diffusion model. Here, we incorporate the warped frames $x_w[:,1]$ as guidance to ensure the generated novel views align with the warped images from input video. The warped frames $x_w[:,:]$ are computed as follows. Given an input video $x[1,:]$, we generate novel views by first estimating a per-frame depth map $D[1,:]$ using a monocular depth estimation model (Piccinelli et al., 2024). This depth information enables depth-based geometric warping, wherein each frame of the input video is unprojected into 3D space and reprojected into a target viewpoint in $p(n) \in \mathcal{P}_N$ where $\mathcal{P}_N$ defines the desired set of camera views. This produces the warped frames:

$$x_w[n,i] = \mathcal{W}\left(x[1,i], D[1,i], p(n), K\right), \quad i = 1, \ldots, F, \tag{1}$$

for $n = 1, \cdots, N$, where $K$ is the intrinsic camera matrix. The warping function $\mathcal{W}(\cdot)$ unprojects each pixel using its estimated depth and reprojects it into the target view. Formally, for each pixel location $r_i$ in the $i$-view, the warped pixel location $r_j$ in the novel-view at the $j$-th camera location is computed as:

$$r_j = KP_{i \to j}D_i(r_i)K^{-1}r_i, \tag{2}$$

where $P_{i \to j}$ is the transformation from the input to the novel-view, and $D_i(r_i)$ is the depth at $r_i$. Since $r_j$ may not align exactly with integer pixel locations, interpolation is applied to assign pixel

---

**Algorithm 1:** Zero4D overall pipeline

---

**Input:** Input video $x[1,:]$, warped views $x_w[:,:]$, masks $m_w[:,:]$, Video diffusion interpolator $I_\theta$

**Output:** Spatio–temporally consistent 4D grid $x_0[:,:] \in \mathbb{R}^{N \times F}$

1 **Stage A — Boundary/Keyframe generation**

2 $x[:,1] \sim p_\theta(x[:,1] \mid x_w[:,1], m_w[:,1], c[1,1])$        `// (a1) left column`

3 $x[N,:] \sim p_\theta(x[N,:] \mid x_w[N,:], m_w[N,:], c[N,1])$     `// (a2) bottom row`

4 **for** $t \leftarrow T$ **to** $0$ **do**                               `// (a3) right column`

5     $x_{t-1}[:,F] \leftarrow I_\theta(x_t[:,F], \sigma_t;\ c[1,F], c[N,F], x_w[:,F])$

6 $c[:,1], c[N,:], c[:,F] \leftarrow \text{Encode}(\{x[:,1], x[N,:], x[:,F]\})$

7 **Stage B — Spatio–temporal bidirectional interpolation**

8 $x_T[:,:] \sim \mathcal{N}(0, I)$

9 **for** $t \leftarrow T$ **to** $1$ **do**

10     **for** $i \leftarrow 1$ **to** $F$ **do**                    `// Camera-axis interpolation`

11        $x_{t-1}[:,i] \leftarrow I_\theta(x_t[:,i], \sigma_t;\ c[1,i], c[N,i], x_w[:,i], m_w[:,i])$

          $x_t[:,i] \leftarrow x_{t-1}[:,i] + \sqrt{\sigma_t^2 - \sigma_{t-1}^2}\, \epsilon$         `// re-noise`

12     **for** $j \leftarrow 1$ **to** $N$ **do**                    `// Time-axis interpolation`

13        $x_{t-1}[j,:] \leftarrow I_\theta(x_t[j,:], \sigma_t;\ c[j,1], c[j,F], x_w[j,:], m_w[j,:])$

14 **return** $x_0[:,:]$

---

values. However, missing regions (e.g., occlusions from depth-based projection) often appear in $x_w$. To address this, we utilize a video diffusion model (Blattmann et al., 2023) parameterized by $\theta$ to inpaint the missing regions and ensure consistency within the 4D video grid. This can be considered as conditional sampling under the condition of the warped image, occlusion mask, and the input video conditioning. For the case of novel-view synthesis at the temporal frame index $j = 1$, this corresponds to

$$x[:,1] \sim p_\theta\left(x[:,1] \mid x_w[:,1], m_w[:,1], c[1,1]\right), \tag{3}$$

where $p_\theta$ corresponds to the conditional distribution from the trained diffusion model, $m_w[:,1]$ is an occlusion mask that identifies missing pixels, and $c[1,1]$ is conditioned embedding vector from $x[1,1]$. The specific details of conditional video diffusion sampling will be described in Section 3.3.

**End view video generation (a2).** Similarly, we can synthesize the end-view video $x[N,:]$ from the generated view $x[N,1]$ utilizing warp-guided diffusion sampling.

$$x[N,:] \sim p_\theta\left(x[N,:] \mid x_w[N,:], m_w[N,:], c[N,1]\right). \tag{4}$$

This process follows the same video sampling approach as first-frame novel-view synthesis; however, it differs in that it synthesizes the video from the final camera position.

**End frame novel-view synthesis (a3).** Finally, we generate video at the end-frame novel-view $x[:,F]$, which constitutes the rightmost column of the 4D grid in Fig. 3(a). Given that we already have $x[1,F]$ from the input video and the synthesized end-view frame $x[N,F]$ derived from $x[N,:]$, we incorporate both images to enhance consistency. To this end, we repurpose a video interpolation method that simultaneously conditions on both $c[1,F]$ and $c[N,F]$ for novel-view synthesis. During interpolation, we further incorporate the warped image and its mask to fully exploit the available prior information. In particular, we synthesize the last column $x[:,F]$ leveraging video diffusion interpolation method (Yang et al., 2025):

$$x_{t-1}[:,F] = I_\theta\left(x_t[:,F], \sigma_t, c[1,F], c[N,F], x_w[:,F]\right) \qquad \text{for} \quad t = T \to 0. \tag{5}$$

where $I_\theta$ denotes the one-step denoising using video interpolation. The final novel-view frame $x[:,F]$ is obtained iteratively by applying $I_\theta$ over diffusion time steps $t = T \to 0$. The detailed implementation of the interpolation process is provided in Algorithm 2 of Appendix A.4.

Input video          Camera orbit controls          Input video          Dolly and transition controls

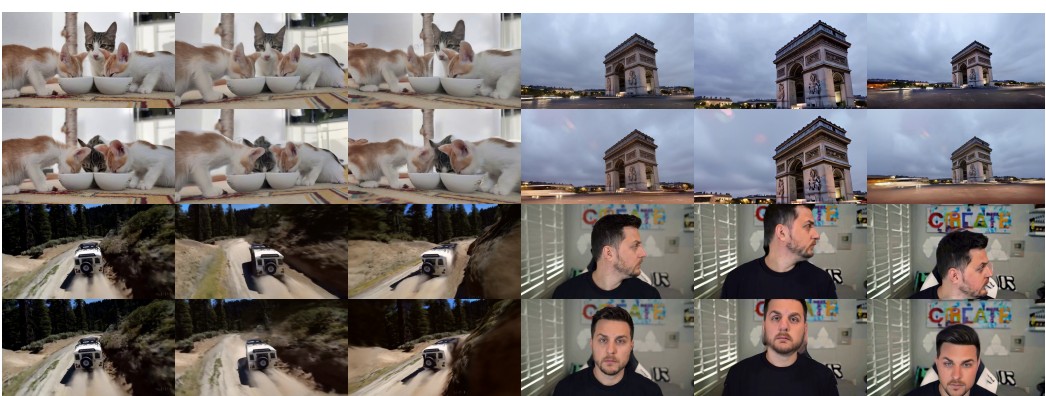

Figure 4: **Result from Zero-4D.** Our model generates high-quality 4D videos from a single input video, enabling diverse camera motions such as orbit, transition, and dolly movements. As illustrated, the synthesized videos maintain spatial and temporal consistency across multiple views and frames, effectively rendering novel perspectives that are not present in the original input. *Best viewed with Acrobat Reader. Click first two rows' images to play the video clip.*

## 3.2 SPATIO-TEMPORAL BIDIRECTIONAL INTERPOLATION

As shown in Fig. 3(b), once the keyframes are generated, the remaining task is to fill in the missing sampling grid at the center so the final resulting 4D video remains consistent across both the camera and time axes. Accordingly, it is essential to perform conditioned sampling using the key frames and adjacent frames from the camera and temporal axes. However, a naive image-to-video diffusion model can only condition on a single or two end frames. To address this challenge, we first repurpose a video interpolation approach to generate spatio-temporally consistent samples under multi-view conditions. The key idea is to alternate interpolation along both the camera and time axes, thereby guiding the overall diffusion trajectory to satisfy the multiple constraints from the keyframes. In this work, we leverage ViBiDSampler (Yang et al., 2025) as the interpolator, with implementation details provided in Appendix A.5 (see Algorithm 2).

**Camera axis interpolation.** Starting from the initial noise $x_T[:,:] \sim \mathcal{N}(0, I)$, we select a specific frame in the 4D grid (a column) $x_t[:,i]$, and perform an interpolation denoising process(6) using the edge-frame conditions $c[1,i]$ and $c[N,i]$:

$$x_{t-1}[:,i] \leftarrow I_\theta(x_t[:,i], \sigma_t, c[1,i], c[N,i], x_w[:,i]) \qquad (6)$$

Here, the image condition $c[1,i]$ is applied first, along with the warped view to guide the diffusion denoising step. The video is then perturbed with noise again, flipped along the camera axis, and subjected to another diffusion denoising step using $c[N,i]$ as the condition. Through these two conditioning steps, $x_t[:,i]$ integrates information from both $c[1,i]$ and $c[N,i]$, enabling interpolation-based denoising that preserves consistency across the camera axis. Before proceeding to time axis interpolation, we apply a re-noising step to ensure smooth transitions across generated frames.

**Time axis interpolation.** After ensuring spatial consistency across the camera axis, we interpolate frames along the time axis to maintain temporal coherence. For each row $x_t[j,:]$ in the 4D grid, we perform an interpolation denoising (7) using the start and end frame conditions $c[j,1]$ and $c[j,F]$.

$$x_{t-1}[j,:] \leftarrow I_\theta(x_t[j,:], \sigma_t, c[j,1], c[j,F], x_w[j,:]) \qquad (7)$$

Initially, $c[j,1]$ is applied along with the warped view to guide the diffusion denoising step. The frame is then perturbed with noise, flipped along the time axis, and another diffusion denoising

step is performed using $c[j, F]$ as the condition. Through this bidirectional conditioning process, $x_t[j, :]$ effectively integrates information from both $c[j, 1]$ and $c[j, F]$, facilitating interpolation-based denoising that ensures smooth transitions along the time axis. Throughout the diffusion steps, we perform denoising by alternating interpolation along the camera axis and time axis. This approach maintains global coherence while ensuring consistency in multi-view video generation.

### 3.3 DETAILS OF CONDITIONAL VIDEO DIFFUSION

Our work is built upon Stable Video Diffusion (SVD) (Blattmann et al., 2023), an image-to-video diffusion model that follows the principles of the EDM framework (Karras et al., 2022). SVD utilizes an iterative denoising approach based on an Euler step method, which progressively transforms a Gaussian noise sample $x_T$ into a clean signal $x_0$:

$$x_{t-1}(x_t; \sigma_t, c) := \hat{x}_c(x_t) + \frac{\sigma_{t-1}}{\sigma_t}\left(x_t - \hat{x}_c(x_t)\right), \qquad (8)$$

where the initial noise is $x_T \sim \mathcal{N}(0, I)$, $\hat{x}_c(x_t)$ is the denoised estimate by Tweedie's formula using the score function trained by the neural network parameterized by $\theta$, and $\sigma_t$ is the discretized noise level for each timestep $t \in [0, T]$.

Now, we describe how to modify SVD to enable conditional sampling under the condition on warped image $x_w$, occlusion mask $m$, and conditioning input $c$. For convenience, we refer to $x_t[:, :]$ as $x_t$. From the formulation of the reverse diffusion sampling process in Eq. (8), the reverse diffusion process can be modulated by conditioning on a known scene-prior $x_{\text{known}}$ (Lugmayr et al., 2022):

$$\bar{x}_c(x_t) = \hat{x}_c(x_t) \cdot m + x_{\text{known}} \cdot (1 - m), \qquad (9)$$

where $m$ is a mask that determines which parts of the scene are known, guiding the denoising process by preserving the warped pixels while allowing the diffusion model to inpaint the missing areas. In our approach, rather than relying on an externally defined scene-prior $x_{\text{known}}$, we leverage the warped frames $x_w$ obtained from depth-based warping as the conditional guidance. Specifically, we redefine the denoising process by replacing $x_{\text{known}}$ with $x_w$ and substituting $m$ with the occlusion mask $m_w$:

$$\bar{x}_c(x_t) = \hat{x}_c(x_t) \cdot m_w + x_w \cdot (1 - m_w). \qquad (10)$$

Here, the occlusion mask $m_w$ ensures that the visible regions in $x_w$ directly guide the denoising process, while the unseen parts are inpainted using the learned prior. By incorporating this modified formulation into the reverse diffusion process, we obtain the following sampling update:

$$x_{t-1}(x_t; \sigma_t, c) \leftarrow \bar{x}_c(x_t) + \frac{\sigma_{t-1}}{\sigma_t}\left(x_t - \hat{x}_c(x_t)\right), \qquad (11)$$

where the target camera viewpoints influence the generated frames through the depth-warped observations $x_w$, ensuring geometric consistency during video synthesis. Throughout the reverse sampling, we iteratively apply this procedure. Additionally, following the approach of (Lugmayr et al., 2022; Liu et al., 2024), we incorporate resampling annealing to further enhance output quality.

## 4 EXPERIMENTS

We used the SVD (Blattmann et al., 2023) as an I2V model without additional training. The image resolution was fixed at 576×1024, with 25 cameras and a sequence length of 25 frames, a total of multi-view video frames are $625 = 25^2$. All frames were generated to form a multi-view video following the target camera trajectory. For depth-based warping, we utilized off-the-shelf depth models (Hu et al., 2024) with various camera movements, including orbit controls (right, left), dolly in/out, and vertical transitions (up, down), with further details on the camera movements provided in Appendix A.3. Runtime performance and user study in appendix A.1 and A.4 confirm that our method is much more memory-efficient and competitive in runtime, outperforming the baselines.

**Baseline models.** We compare against state-of-the-art video generation models that support either camera control or multi-view generation: (1) CameraCtrl (He et al., 2024) is a camera-controllable video diffusion model. Given a single input image, it can synthesize bullet-time videos by following a predefined camera trajectory. (2) TrajectoryCrafter (YU et al., 2025), a representative baseline, synthesizes novel-view and bullet-time videos from warped frames aligned to a target trajectory. (3) TrajectoryAttention (Xiao et al., 2025) similarly leverages warped video frames from the input video to generate both novel-view and bullet-time videos. (4) SV4D (Xie et al., 2024) is an image-to-video

Table 2: **Quantitative result in novel view video generation.** We evaluate our method against baselines on VBench, comparing multi-view video results based on novel-view generation from a fixed camera view. Our method achieves the best performance in both frame consistency across videos and image quality of individual frames. (* denotes baselines evaluated with bullet-time re-alignment)

| Method | Subject Consistency ↑ | Background Consistency ↑ | Temporal Flickering ↑ | Motion Smoothness ↑ | Dynamic Degree ↓ | Image Quality ↑ | Aesthetic Quality ↑ |
|---|---|---|---|---|---|---|---|
| SV4D | 88.76% | 91.36% | 94.21% | 95.28% | 49.20% | 46.89% | 34.36% |
| GCD | 90.31% | 94.13% | 96.14% | 93.21% | 19.23% | 45.77% | 32.98% |
| TrajectoryAttention | 88.83% | 91.42% | 96.86% | 97.89% | 59.50% | 42.98% | 37.92% |
| TrajectoryCrafter | 93.47% | **96.93%** | **98.42%** | **99.26%** | 21.00% | **52.10%** | **44.41%** |
| Ours | **95.55%** | 95.75% | 97.48% | 98.34% | 27.50% | 51.12% | 38.22% |
| CameraCtrl* | 91.71% | 91.05% | 89.98% | 91.03% | 98.00% | 40.12% | 35.86% |
| TrajectoryAttention* | 94.72% | 94.93% | **97.61%** | 98.28% | **27.50%** | 47.75% | **42.88%** |
| TrajectoryCrafter* | 94.71% | 94.48% | 94.74% | 96.81% | 32.50% | 48.81% | 35.86% |
| Ours | **95.55%** | **95.75%** | 97.48% | **98.34%** | **27.50%** | **51.12%** | 38.22% |

diffusion model capable of generating multiple novel-view videos from a single input video. (5) GCD (Van Hoorick et al., 2024a) also takes a single video as input and generates novel views of dynamic 4D scenes by controlling azimuth and elevation angles.

**Evaluation protocol.** We evaluate our method in two categories: (1) fixed novel-view video generation and (2) bullet-time video generation. For novel-view evaluation, we adopt VBench (Huang et al., 2024), which measures seven aspects of video quality, including identity retention, motion coherence, and temporal consistency. For bullet-time evaluation, we assess 3D consistency using pose errors (ATE, RPE-T, RPE-R) (Goel et al., 1999) obtained via COLMAP (Schönberger & Frahm, 2016) and MEt3R (Asim et al., 2024), a recent metric based on DUSt3R (Wang et al., 2024b) that quantifies geometric consistency from unposed frames. We conducted all experiments on 50 videos randomly sampled from Webvid-10M (Bain et al., 2021), comparing ours with baseline models.

### 4.1 FIXED NOVEL-VIEW VIDEO GENERATION

We evaluate our method in two settings: (1) novel-view generation for video quality, and (2) spatio-temporal consistency for coherence across views and time.

**Evaluation of direct novel-view generation.** We assess the quality of novel-view videos from fixed target viewpoints using VBench (Huang et al., 2024). Zero4D retrieves $x[n,:]$ corresponding to a target camera viewpoint $p(n)$ from the 4D video grid $x[:,:]$ synthesized from the input video $x[1,:]$, while baselines directly generate $x[n,:]$ at viewpoint $p(n)$. For this experiment, we consider baselines capable of direct novel-view generation at viewpoint $n$, SV4D, GCD, TrajectoryAttention, and TrajectoryCrafter. As shown in the upper part of Table 2, Zero4D, despite being training-free, achieves the highest score in subject consistency and ranks second in five other categories. This demonstrates that ours achieves robust novel-view video generation performance, comparable to models pretrained on large-scale datasets.

**Evaluation of global spatio-temporal consistency.** To examine whether models maintain global 4D consistency, we construct re-aligned videos at a fixed viewpoint from generated bullet-time videos. For each input frame $x[1,i]$ ($i = 1, \ldots, F$), baselines generate a bullet-time sequence $x[:,i]$ along a predefined trajectory. These sequences are aggregated into a 4D grid $x[:,:]$, from which the fixed-view sequence $x[n,:]$ at viewpoint $p(n)$ is extracted. We consider three baseline models capable of bullet-time video generation: CameraCtrl, TrajectoryAttention, and TrajectoryCrafter. In contrast, Zero4D directly retrieves $x[n,:]$ from its generated 4D grid without requiring bullet-time re-alignment. As shown in the Table 2 (below the horizontal separator), ours achieves the highest scores in five VBench categories and second-best in the remaining two. This strong performance indicates that spatio-temporal interpolation enables Zero4D to preserve global consistency across views and time, whereas baseline models, unable to sample jointly across multi-view and multi-time dimensions, yield inferior consistency. Although baseline models generate plausible bullet-time results at individual time steps, re-alignment to a fixed viewpoint exposes frequent inconsistencies, particularly in the background and the x–t slices shown in Figure 2, which clearly reveal the inconsistencies.

### 4.2 BULLET-TIME VIDEO GENERATION

We design two evaluations for bullet-time video generation: (1) direct generation along a camera trajectory to assess spatial coherence, and (2) multi-view alignment at fixed time steps to measure global 4D consistency.

**Evaluation of direct bullet-time generation.** We compare Zero4D against baselines (CameraCtrl, TrajectoryAttention, TrajectoryCrafter) capable of bullet-time generation. Given an input video $x[1, :]$, these models generate bullet-time sequences $x[:, i]$ by smoothly moving the camera along a predefined trajectory at fixed time $i$. This setting provides a direct evaluation of each model's ability to produce spatially coherent bullet-time videos from the input. As shown in Table 3 (upper part), Zero4D attains comparable scores to baselines that are explicitly trained for novel-view video generation, despite being a training-free approach.

**Evaluation of multi-view consistency in bullet-time.** To further assess global 4D consistency, we construct bullet-time videos by re-aligning novel-view outputs at a fixed time step. For baseline models, novel-view videos $x[n, :]$ are generated at each target viewpoint $p(n)$ along the predefined camera trajectory, and the frames corresponding to the same time index are re-aligned to form a bullet-time sequence $x[:, :]$. In contrast, ours directly retrieves the corresponding sequence $x[:, i]$ from its

Table 3: **Bullet-time video quantitative comparisons.** We report results on (1) direct bullet-time generation for spatial coherence and (2) multi-view consistency by re-aligning outputs at fixed time steps.(* denotes baselines evaluated with novel-view re-alignment)

| Method | ATE (m, ↓) | RPE-T (↓) | RPE-R (deg ↓) | MEt3R ↓ |
|---|---|---|---|---|
| CameraCtrl | 0.185 | 0.155 | 0.57 | 0.0264 |
| TrajectoryAttention | 0.182 | **0.113** | **0.25** | **0.0202** |
| TrajectoryCrafter | **0.170** | 0.140 | 2.26 | 0.0224 |
| Ours | 0.190 | 0.142 | 0.53 | 0.0307 |
| TrajectoryAttention* | 5.582 | 3.377 | 1.65 | 0.1000 |
| TrajectoryCrafter* | 0.211 | 0.251 | 3.61 | 0.0930 |
| Ours | **0.190** | **0.142** | **0.53** | **0.0307** |

generated 4D grid $x[:, :]$, without requiring re-alignment. As shown in Table 3 (below the horizontal separator), Zero4D maintains global coherence across views and time, thereby achieving better accuracy in pose estimation (ATE, RPE-T, RPE-R) and lower MEt3R scores, surpassing the performance of baseline approaches.

Table 4: **Quantitative ablation.** Ablation studies on generated videos show that incorporating all components yields the best performance.

| Method | ATE (m,↓) | RPE-T (m, ↓) | RPE-R (deg, ↓) | Subject Consistency ↑ | Background Consistency ↑ | Temporal Flickering ↑ | Motion Smoothness ↑ | Dynamic Degree ↓ | Image Quality ↑ | Aesthetic Quality ↑ |
|---|---|---|---|---|---|---|---|---|---|---|
| Ours | 0.190 | **0.142** | 0.53 | **95.55%** | **95.75%** | **97.48%** | **98.34%** | 27.50% | 51.12% | 38.22% |
| w/o STBI | **0.175** | 0.149 | **0.34** | 93.23% | 92.63% | 93.28% | 95.24% | 100% | **52.38%** | **43.21%** |
| w/o warp | 0.501 | 0.251 | 0.89 | 93.73% | 93.38% | 93.98% | 96.12% | 47.29% | 43.79% | 36.11% |

**Ablation.** We performed ablation studies under two settings: (1) *Without warped frame guidance:* removing warped frames from the input degrades image fidelity and weakens structural details. (2)*Without spatio-temporal bidirectional interpolation (STBI):* generating each novel-view independently breaks multi-view coherence. Table 4, evaluated with ATE, RPE-T, RPE-R in the bullet-time setting and VBench (Huang et al., 2024) for fixed novel-view, shows that both components are essential for maintaining fidelity and global consistency. Additional qualitative ablation results are provided in Appendix A.5 (see Figure 7).

## 5 CONCLUSION

In this work, we introduced a novel training-free approach for synchronized multi-view video generation using an off-the-shelf video diffusion model. Our method generates high-quality 4D video through depth-based warping and spatio-temporal bidirectional interpolation, ensuring structural consistency across both spatial and temporal domains. Unlike prior methods that rely on extensive training with video or 4D datasets, our framework achieves competitive performance without additional training. Experiments demonstrate that our approach produces synchronized multi-view videos with superior subject consistency, smooth motion trajectories, and temporal stability. This makes our framework a practical solution for multi-view video generation, particularly in scenarios where large-scale 4D datasets and powerful computational resources are limited. Future work may investigate extensions to more complex dynamic scenes, adaptive interpolation strategies, or fusion with other generative models to further enhance realism and flexibility.

**Limitation and Potential Negative Impacts.** While our method enables training-free 4D generation, it requires multiple rounds of bidirectional diffusion sampling, which leads to increased inference time. Additionally, since the 4D generation is guided by the prior knowledge encoded in the pre-trained video diffusion model, our method may inherit potential drawbacks of generative models.

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

# A APPENDIX

## A.1 USER STUDY.

To evaluate our approach, we conducted a user study comparing Ours, TrajectoryCrafter (YU et al., 2025), TrajectoryAttention (Xiao et al., 2025), and CameraCtrl (He et al., 2024) across four key metrics: View Angle, General Quality, Smoothness, and Background Quality. Participants viewed generated videos and selected the most visually appealing results for each criterion, providing subjective

Table 5: **User study.** Winning rates across four evaluation metrics. Our method consistently outperforms the baselines, particularly in General Quality and Background Quality.

| Method | View Angle | General Quality | Smoothness | BG Quality |
|---|---|---|---|---|
| Ours | 30% | **36%** | 33% | **39%** |
| TrajectoryCrafter | **32%** | 30% | 27% | 28% |
| TrajectoryAttention | 27% | 26% | **34%** | 23% |
| CameraCtrl | 11% | 8% | 6% | 10% |

feedback on the overall quality and realism. As shown in Table 5, our method consistently achieved the highest user preference, particularly excelling in General Quality (36%) and Background Quality (39%), which highlights its superior fidelity and ability to preserve scene details. The View Angle metric (30%) confirms accurate and convincing novel-view synthesis, while Smoothness (33%) indicates our approach produces fluid transitions with minimal distortion or artifacts. These results collectively demonstrate that our method offers a more immersive and visually coherent experience compared to competing techniques.

## A.2 PRE-TRAINED MODEL CHECKPOINTS

Zero4D is developed based on publicly available, pre-trained generative models for both images and videos. For transparency and reproducibility, we specify below the exact versions of each model employed in our framework:

- Depth estimation model: Depthcrafter [1]
- Image-to-Video generation model: stable-video-diffusion-img2vid-xt[2]

## A.3 CAMERA TRAJECTORY CONTROL

We support various camera motions for novel view synthesis, leveraging depth information for realistic scene transformation:

**Camera orbit rotation:** Horizontal camera movement around the subject, creating a side-to-side viewing effect. The depth map guides proper parallax by determining each pixel's displacement based on its relative depth.

**Dolly movement:** Forward/backward camera translation that adjusts focal length to maintain subject size. For dolly-in, foreground elements remain stable while the background compresses; for dolly-out, the background expands naturally.

**Elevation transition:** Vertical camera movement that rotates the viewpoint up or down. Depth information ensures accurate perspective shifts as the camera changes height, maintaining geometric consistency.

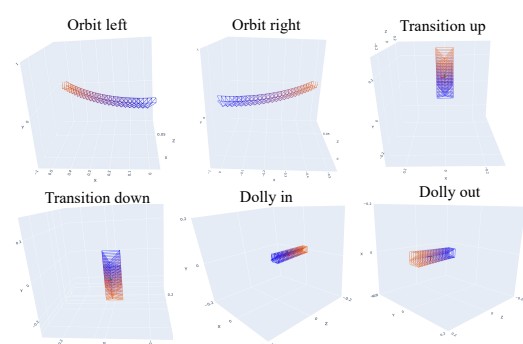

Figure 5: **Camera trajectory visualization.** With a monocular depth estimation model, our approach can generate various novel view videos with spatio-temporal synchronized videos.

Our system utilizes monocular depth estimation to construct a pseudo-3D representation of the scene. This depth map is crucial for maintaining geometric consistency during novel view synthesis, allowing for convincing parallax effects and occlusion handling. By projecting pixels according to their estimated depth values, we achieve realistic scene transformations without explicit 3D reconstruction.

---

[1]https://huggingface.co/tencent/DepthCrafter
[2]https://huggingface.co/stabilityai/stable-video-diffusion-img2vid-xt

## A.4 DETAILS OF ZERO4D IMPLEMENTATION

---

**Algorithm 2:** $I_\theta$ : A sampling step of extended ViBiDSampler for bidirectional interpolation.

1: **function** $I_\theta(x_t, \sigma_t, c_{start}, c_{end}, x_w)$
2:      $\hat{x}_{c_{start}} \leftarrow D_\theta(x_t; \sigma_t, c_{start})$         $\triangleright$ EDM denosing
3:      $\bar{x}_{c_{start}} \leftarrow \hat{x}_{c_{start}} \cdot m + x_w \cdot (1-m)$
4:      $x_{t-1,c_{start}} \leftarrow \bar{x}_{c_{start}} + \frac{\sigma_{t-1}}{\sigma_t}(x_t - \hat{x}_\emptyset)$
5:      $x_t, c_{start} \leftarrow x_{t-1,c_{start}} + \sqrt{\sigma_t^2 - \sigma_{t-1}^2}\epsilon$         $\triangleright$ Re-noise
6:      $x_t, c_{start} \leftarrow \text{flip}(x_t, c_{start})$         $\triangleright$ Time reverse
7:      $\hat{x}'_{c_{end}} \leftarrow D_\theta(x'_t, c_{start}; \sigma_t, c_{end})$         $\triangleright$ EDM denoising
8:      $\bar{x}'_{c_{end}} \leftarrow \bar{x}'_{c_{end}} \cdot m + x_w \cdot (1-m)$
9:      $x'_{t-1} \leftarrow \bar{x}'_{c_{end}} + \frac{\sigma_{t-1}}{\sigma_t}(x'_t, c_{start} - \hat{x}'_\emptyset)$
10:     $x'_{t-1} \leftarrow \text{flip}(x'_{t-1})$         $\triangleright$ Time reverse
11:     **return** $x_{t-1}$
12: **end function**

---

**Algorithm 3:** Novel view synthesis and end-view video generation algorithm from Liu et al. (2024)

---

**Input:** Warped frames $x_w$, opacity mask $m$
**Output:** Input video $x_0$
1   $x_T \sim \mathcal{N}(0,1)$
2   **for** $t \leftarrow T$ **to** $1$ **do**
3      **if** $t > T - T^{guide}$ **then**
4          **for** $r \leftarrow 1$ **to** $R$ **do**
5             $\hat{x}_0 \leftarrow \text{Predict}(x_t)$
6             **if** $r \leq R^{guide}$ **then**
7                $\hat{x}_0 \leftarrow D_\theta(x_t; \sigma_t, c_{x_0})$
8                $\bar{x}_0 \leftarrow \hat{x}_0 \cdot m + x_w \cdot (1-m)$
9             **else**
10               $\bar{x}_0 \leftarrow \hat{x}_0$
11             $x_{t-1} \leftarrow \bar{x}_0 + \frac{\sigma_{t-1}}{\sigma_t}(x_t - \hat{x}_0)$
12             **if** $r < R$ **then**
13               $x_t \sim \mathcal{N}(\bar{x}_0, \sigma_t)$
14      **else**
15          $\hat{x}_{t-1} \leftarrow D_\theta(x_t; \sigma_t, c_{x_0})$
16          $x_{t-1} \leftarrow \bar{x}_0 + \frac{\sigma_{t-1}}{\sigma_t}(x_t - \hat{x}_0)$
17   **return** $x_0$

---

**Details of interpolation.** To generate globally consistent 4D videos, we adapt the interpolation strategy during spatio-temporal video generation. Specifically, we leverage ViBiDSampler (Yang et al., 2025) as the interpolator $I_\theta$. ViBiDSampler is a state-of-the-art training-free video interpolation method designed for image-to-video diffusion models. Given two conditioning frames, it alternates denoising along the temporal axis to synthesize intermediate frames. In our framework, we extend this process by incorporating warped-frame guidance (see Algorithm 2), which provides additional geometric cues. This modification refines the interpolation process, leading to more faithful structure preservation and improved global spatio-temporal coherence across the generated 4D video grid.

**Novle-view synthesis.** Algorithm 3 outlines the process for generating novel-view videos from a single monocular video. We first apply novel view synthesis to the initial frame using an I2V diffusion model Blattmann et al. (2023) to produce the novel view $x[:, 1]$. For this, depth-based warping priors from the input video are incorporated to enable inpainting-based synthesis. Specifically, using an off-the-shelf depth estimation model Piccinelli et al. (2024); Hu et al. (2024), we warp the original

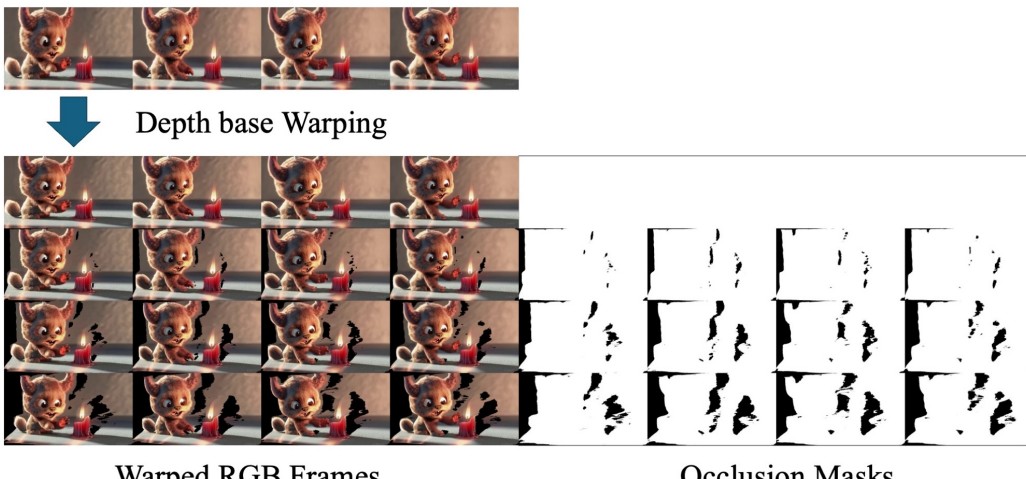

Figure 6: **Input Video Warping.** Given a single video, we utilize an off-the-shelf depth estimation model to generate warped frames from novel viewpoints.

frame to novel viewpoints, as illustrated in Figure 5. As shown in Fig. 6, occluded regions from the warp operation appear black, allowing us to extract an opacity mask. Inspired by Lugmayr et al. (2022); You et al. (2024); Liu et al. (2024), we adopt a mask inpainting approach, where inpainting is performed on the estimated noisy frame $\hat{x}_0[:, 1]$. Rather than applying inpainting at every denoising step, as in Liu et al. (2024), we utilize a re-noising process within the diffusion model's denoising step to refine the final synthesis by reducing artifacts and enhancing structural coherence. A detailed description is provided in Algorithm 3.

Table 6: Runtime and VRAM comparison for sampling an $N \times F$ 4D grid.

|  | Zero4D (Ours)* | Zero4D (Ours) | TrajectoryCrafter | TrajectoryAttention | CameraCtrl |
|---|---|---|---|---|---|
| VRAM | 23GB | 28GB | 45GB | 20GB | 46GB |
| Time | 88m | 66m | 60m | 50m | 31m |

**Runtime performance.** Although Zero4D takes a similar amount of time to generate a full 4D video compared to baseline methods, it requires significantly less GPU memory—nearly 40–50% less than TrajectoryCrafter. This makes Zero4D a much more memory-efficient solution that remains competitive in runtime without compromising consistency or quality. The * indicates results measured on an RTX 4090, while the others were benchmarked on an NVIDIA A100.

## A.5 Additional Results

**Ablation (detailed analysis).**

Figure 7 qualitatively illustrates the role of each component in maintaining global consistency. Without spatio-temporal bidirectional interpolation (STBI), each frame is synthesized independently, which causes temporal flickering and background inconsistencies across views. For example, in the water-pouring sequence (left), the liquid surface fails to remain temporally stable, as highlighted by the red boxes. Similarly, without warping guidance, the model struggles with geometric alignment. In the motorcycle example (middle), artifacts appear in the generated human figure, leading to distorted or incomplete shapes. Finally, in the clock sequence (right), the absence of warping or spatio-temporal interpolation leads to visible structural mismatches and background inconsistencies. In contrast, our full model effectively aggregates global information through STBI and enforces geometric consistency via warped-frame guidance, resulting in coherent and high-quality multi-view videos across both spatial and temporal dimensions.

Figure 7: **Ablation results.** Removing spatio-temporal bidirectional interpolation (STBI) or warping guidance leads to broken consistency and geometric artifacts (red boxes). In contrast, our full method preserves spatial structure and temporal coherence across views.

## B   THE USE OF LARGE LANGUAGE MODELS (LLMs)

LLMs were not involved in research ideation or methodological design and were only used for minor expression refinement. The authors retain full responsibility for all scientific content.

Input video    View from orbit left    View from orbit right

Input video    Transition up    Transition down

Figure 8: **Camera orbit & transition novel videos.** Our model generates high-quality 4D videos from a single input video, enabling diverse camera motions such as orbit, transition, and dolly movements. *Best viewed with Acrobat Reader. Click the images to play the video clip.*

Input video                          Time index 1                          Time index 2

Figure 9: **Bullet time videos.** Our model generates high-quality bullet-time videos, demonstrating spatio-temporal consistency. *Best viewed with Acrobat Reader. Click the images to play the video clip.*

Input video          Dolly in          Dolly out

Static view

Dynamic view

Static view

Dynamic view

Static view

Dynamic view

Static view

Dynamic view

Figure 10: **Dolly in/out videos.** Our model generates high-quality 4D videos from a single input video with dolly movements. *Best viewed with Acrobat Reader. Click the images to play the video clip.*