# OpenReview forum: "Zero4D: Training-Free 4D Video Generation From Single Video Using Off-the-Shelf Video Diffusion Models"
_ICLR.cc/2026/Conference — ICLR 2026 Conference Withdrawn Submission_

### Official Review · Reviewer_gfab · 2025-10-30

**Soundness:** 3
**Presentation:** 3
**Contribution:** 3
**Rating:** 6
**Confidence:** 4

**Summary:**

This paper proposes a training-free method to generate synchronized multi-view videos from a single input video by leveraging an off-the-shelf video diffusion model.

It generates the edge frames in the spatial-temporal image matrices as key frames. These edge frames are generated using a video diffusion model, guided by depth-based warping.

Then it interpolates the remaining frames using a video diffusion model again.

The framework outperforms baselines in maintaining spatial-temporal consistency and 4D generation.

**Strengths:**

1. The paper is well-written and easy to understand.
2. The proposed method is interesting and novel.
3. The proposed framework achieves comparable results to other training-based methods.

**Weaknesses:**

1. The depth-warping strategy inherently limits the range of applicable camera viewpoints. When the baseline between views becomes large, severe occlusions may occur, potentially causing the guided generation process to fail. It would be helpful for the authors to discuss this limitation and clarify how their method handles or mitigates such cases.

2. The paper does not include a comparison with DimensionX [1], which appears to be a highly relevant and recent work. Including this baseline or explaining the rationale for its exclusion would strengthen the evaluation.

3. Missing references: ReCamMaster [2] and SynCamMaster [3] should be cited and compared.

[1] Sun, Wenqiang, et al. "Dimensionx: Create any 3d and 4d scenes from a single image with controllable video diffusion." arXiv preprint arXiv:2411.04928 (2024).
[2] Bai, Jianhong, et al. "Recammaster: Camera-controlled generative rendering from a single video." arXiv preprint arXiv:2503.11647 (2025).
[3] Bai, Jianhong, et al. "Syncammaster: Synchronizing multi-camera video generation from diverse viewpoints." arXiv preprint arXiv:2412.07760 (2024).

**Questions:**

1. If the goal is to generate more than 25 frames and 25 views, how can the proposed approach be scaled or extended to handle longer temporal sequences or a larger number of viewpoints? A discussion of potential limitations or solutions in this regard would be valuable.

2. Since self-occlusion can occur in the input view, there may be inconsistencies between the input video frames (first-row images) and the generated multi-view outputs (first-column images). Could this issue become significant in certain scenarios? It would be helpful for the authors to elaborate on how their method addresses or mitigates this potential problem.

---

### Official Review · Reviewer_NvsT · 2025-10-31

**Soundness:** 3
**Presentation:** 3
**Contribution:** 3
**Rating:** 4
**Confidence:** 4

**Summary:**

This paper introduces a novel, training-free method for generating 4D videos (multi-view videos over time) from a single input video. The core innovation is a framework that cleverly uses existing, off-the-shelf video diffusion models without requiring any additional training or access to scarce real-world 4D data.

**Strengths:**

1. This paper proposes a training-free 4D video generation method that leverages the priors of existing video diffusion models to ensure spatio-temporal consistency in the generated results.
2. By integrating keyframe generation with spatio-temporal bidirectional interpolation, the method achieves better spatial-temporal consistency than TrajectoryCrafter.
2. Compared to previous 4D video generation approaches, this method avoids the limitation of scarce 4D data, offering improved practicality.

**Weaknesses:**

1. The method relies on depth information for geometric warping, which limits its ability to handle large-angle novel view generation. In contrast, training-based methods such as CAT4D and DimensionX can generate 360° rotational views.
2. Since warped frames are used as conditional inputs (as shown in Eq. (10)), the generated regions filled by the video diffusion model exhibit clear seams or boundaries (e.g., the clock sequence in Figure 9), degrading the overall visual quality.
3. During keyframe generation, only a small number (1 or 2) of images are used as conditions each time. The paper does not sufficiently explain how spatio-temporal consistency among keyframes is maintained. If inconsistencies occur between keyframes, would they propagate and degrade the quality of the final 4D video?
4. The proposed Spatio-Temporal Bidirectional Interpolation is conceptually similar to the multi-loop refinement in DimensionX, thus lacking novelty.
5. The experimental comparisons are insufficient. The paper mainly compares with baselines that generate single-row or single-column videos, but lacks comprehensive comparisons with direct 4D generation methods such as DimensionX.
6. In Table 4, removing the STBI module leads to lower ATE and RPE-R values, as well as higher image quality and aesthetic quality. This suggests that the proposed method may trade off geometric accuracy and visual fidelity for improved spatio-temporal consistency. Furthermore, it is unclear why a lower “Dynamic Degree” value indicates better performance.

**Questions:**

see weaknesses

---

### Official Review · Reviewer_fasy · 2025-11-02

**Soundness:** 3
**Presentation:** 2
**Contribution:** 3
**Rating:** 2
**Confidence:** 4

**Summary:**

This proposes a training-free 4D video generation method from a single monocular video. It consists of two stages. First, an off-the-shelf video diffusion model is used to generate video frames on the boundaries of the 4D image grid. Second, another off-the-shelf video frame interpolation method is adopted to synthesize the in-between frames in the 4D grid.

**Strengths:**

1. The training-free nature of this approach is appealing as the training data of 4D generation is essentially very limited especially on the scene level. Furthermore, it bypasses the computational burden of fine-tuning models.

2. Presentation of the technical content, although can be further improved, is clear and easy to follow.

**Weaknesses:**

1. My major concern of the proposed approach is the warping part. It is notoriously known that such progressive waping-based approach can not generate high-quality videos with large viewpoint change as more and more artifacts will be accumulated. The results shown in this paper seem only from limited camera motion trajectories (e.g., Fig. 5 in Section A.3).

2. The warping module relies on depth estimation. For the depth part, it is essentially based on monocular cues in each frame and thus prune to the inconsistent scales across different frames. For the video generation (a2) part, it also involves camera transformation/pose estimation. It is not clear how the camera poses are obtained.

3. There is not supplementary video submitted. Making it hard to gauge the effectiveness of the proposed approach. The sampled video frames provided in the paper and appendix are very limited.

**Questions:**

1. In Section A.2, it is mentioned that DepthCrafter is used for depth estimation. But in Section 3.1, it is mentioned that UniDepth is adopted. Could you please clarify?

2. Is it possible to report image-quality related metrics, like PSNR, SSIM, for the generated results.

---

### Official Review · Reviewer_Avbg · 2025-11-03

**Soundness:** 2
**Presentation:** 2
**Contribution:** 3
**Rating:** 4
**Confidence:** 4

**Summary:**

This paper introduces Zero4D, a novel framework for generating 4D videos from a single video input. Its primary and most significant contribution is that the method is entirely training-free, cleverly repurposing off-the-shelf video diffusion models to accomplish this complex task.
The proposed approach consists of two main stages: Key Frame Generation and Spatio-Temporal Bidirectional Interpolation (STBI).
By designing the "boundary frames" of a spatio-temporal sampling grid as key frames. It then synthesizes these frames using a video diffusion model, guided by depth-based warping to ensure structural and temporal consistency. After the boundary keyframes are established, the framework leverages the video diffusion model's interpolation capabilities. Experimental results demonstrate that, without any additional training or access to multi-view data, the proposed method claims superior performance in maintaining global spatio-temporal consistency compared to existing baseline models.

**Strengths:**

1. Training-free: They propose a novel framework to achieve training-free 4D video generation from a single video. This is a major paradigm shift, as current leading methods rely on computationally expensive training with large-scale, yet scarce, 4D datasets.

2. Clear Methodology: The decomposition of the complex 4D generation task into "key frame generation" and "spatio-temporal interpolation" is clear and easy to understand. By alternating interpolation across the view and time axes, STBI forces the diffusion sampling process to align with multiple constraints, thus ensuring global consistency. It is a good repurposing of existing model capabilities to solve a higher-dimensional problem.

**Weaknesses:**

1. The authors claim Zero4D is a training-free 4D video generation method that leverages off-the-shelf video diffusion models, while just one I2V model is evaluated in the EXPERIMENTS.

2. Although the author's bi-interpolation strategy can predict the entire view-time grid, it is an inefficient approach. This is because the actual requirement is often just a single trajectory, rendering the computation of the full grid unnecessary and wasteful.

3. Although STBI is designed to enforce global consistency, the iterative nature of filling the grid via alternating interpolation may still result in error accumulation. When generating wider camera trajectories or longer video sequences, minor inconsistencies or artifacts introduced in early interpolation steps could potentially be propagated and amplified in subsequent steps.

**Questions:**

1. When evaluating the global spatial-temporal consistency between Zero4D and other baselines, how do you do re-alignment for baseline models? Because this may somehow affect the final result. Thus, it may be unfair to evaluate these models when considering “FIXED NOVEL-VIEW VIDEO GENERATION ”

2. How does Zero4D 's performance (both in quality and runtime) scale with the dimensions of the view-time grid (i.e., number of cameras N and number of frames F)? The paper experiments with a 25x25 grid. If the video duration is extended significantly (e.g., to 100+ frames) or the camera trajectory is made wider, can the STBI process still effectively mitigate error accumulation and maintain global consistency?

3. The author uses SVD as the baseline model, which supports 3D view generation, while the views of the demo video in this paper have limited degrees of freedom. What is the maximum degree of freedom that Zero4D can support? Can you give some more free-form examples and compare with SVD and other 4D models like SV4D?

@article{xie2024sv4d, title={{SV4D}: Dynamic 3D Content Generation with Multi-Frame and Multi-View Consistency}, author={Yiming Xie and Chun-Han Yao and Vikram Voleti and Huaizu Jiang and Varun Jampani}, journal={arXiv preprint arXiv:2407.17470}, year={2024},}

4. Can Zero4D be adapted to other I2V models, such as Wan2.1-I2V?

5. For a more complete understanding of the method's limitations, a discussion or visualization of typical failure cases would be highly beneficial. For instance, what are the characteristic artifacts in the final 4D video when the initial depth estimation is poor, or when the base diffusion model fails to plausibly inpaint a warped region? This would be crucial for understanding the robustness of the framework.

---

### Note · Authors · 2025-11-14

I have read and agree with the venue's withdrawal policy on behalf of myself and my co-authors.